# Making inroads to precision medicine for the treatment of autoimmune diseases: Harnessing genomic studies to better diagnose and treat complex disorders

Autoimmunity; bioinformatics; data integration; diagnosis; genomics

**Corresponding author:**
Marina Sirota;
Email: Marina.Sirota@ucsf.edu

Yuriy Baglaenko[1], Catriona Wagner[2], Vijay G. Bhoj[3], Petter Brodin[4], M. Eric Gershwin[5], Daniel Graham[6], Pietro Invernizzi[7,8], Kenneth K. Kidd[9], Ilya Korsunsky[1], Michael Levy[10], Andrew L. Mammen[11], Victor Nizet[12], Francisco Ramirez-Valle[13], Edward C. Stites[9], Marc S. Williams[14], Michael Wilson[15], Noel R. Rose[2], Virginia Ladd[2] and Marina Sirota[16,17]

[1]Brigham and Women's Hospital, Boston, MA, USA; [2]Autoimmune Association, Clinton Township, MI, USA; [3]University of Pennsylvania, Philadelphia, PA, USA; [4]Karolinska Institute, Solna, Sweden; [5]University of California, Davis, Davis, CA, USA; [6]The Broad Institute of MIT and Harvard, Cambridge, MA, USA; [7]Division of Gastroenterology, Center for Autoimmune Liver Diseases, Department of Medicine and Surgery, University of Milano-Bicocca, Monza, Italy; [8]European Reference Network on Hepatological Diseases (ERN RARE-LIVER), IRCCS Fondazione San Gerardo dei Tintori, Monza, Italy; [9]Department of Laboratory Medicine, Yale School of Medicine, New Haven, CT, USA; [10]Massachusetts General Hospital and Harvard Medical School, Boston, MA, USA; [11]National Institute of Arthritis and Musculoskeletal and Skin Diseases, National Institutes of Health, USA; [12]School of Medicine, University of California San Diego, San Diego, CA, USA; [13]Bristol Myers Squibb, New York, NY, USA; [14]Geisinger, Danville, PA, USA; [15]Weill Institute for Neurosciences, Department of Neurology, UCSF, San Francisco, CA, USA; [16]Bakar Computational Health Sciences Institute, UCSF, San Francisco, CA, USA and [17]Department of Pediatrics, UCSF, San Francisco, CA, USA

## Abstract

Precision Medicine is an emerging approach for disease treatment and prevention that takes into account individual variability in genes, environment, and lifestyle. Autoimmune diseases are those in which the body's natural defense system loses discriminating power between its own cells and foreign cells, causing the body to mistakenly attack healthy tissues. These conditions are very heterogeneous in their presentation and therefore difficult to diagnose and treat. Achieving precision medicine in autoimmune diseases has been challenging due to the complex etiologies of these conditions, involving an interplay between genetic, epigenetic, and environmental factors. However, recent technological and computational advances in molecular profiling have helped identify patient subtypes and molecular pathways which can be used to improve diagnostics and therapeutics. This review discusses the current understanding of the disease mechanisms, heterogeneity, and pathogenic autoantigens in autoimmune diseases gained from genomic and transcriptomic studies and highlights how these findings can be applied to better understand disease heterogeneity in the context of disease diagnostics and therapeutics.

## Impact statement

Precision medicine is an emerging approach for disease treatment and prevention that takes into account individual variability in genes, environment, and lifestyle. As defined by Christensen et al. ([2009], *The Innovator's Prescription: A Disruptive Solution for Health Care*), precision medicine is provision of care for diseases that can be precisely diagnosed, whose causes are understood, and which consequently can be treated with rules-based therapies that are predictably effective. Autoimmune diseases are those in which the body's natural defense system loses discriminating power between its own cells and foreign cells, causing the body to mistakenly attack healthy tissues. There are more than 80 types of autoimmune diseases that affect a wide range of organ systems. These conditions are very heterogeneous in their presentation and therefore difficult to diagnose and treat. Achieving precision medicine in autoimmune diseases has been challenging due to the complex etiologies of these conditions, involving an interplay between genetic, epigenetic, and environmental factors. However, recent technological and computational advances in molecular profiling have helped to identify patient subtypes and molecular pathways that can be used to improve diagnostics and therapeutics. This review discusses the current understanding of the disease mechanisms, heterogeneity, and pathogenic autoantigens in autoimmune diseases gained from genomic and transcriptomic studies and highlights how these findings can be applied to better understand disease heterogeneity. Within that framework, improved diagnostics and targeted therapeutic approaches may advance toward precision clinical care of patients with autoimmune diseases.



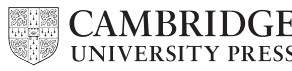

## Introduction

Autoimmune diseases are a diverse group of over 80 diseases, including rheumatoid arthritis (RA), systemic lupus erythematosus (SLE), multiple sclerosis (MS), ulcerative colitis (UC), and many others where the immune system attacks the body. While these diseases are primarily differentiated based on the primary target organ, they also share common features, including loss of tolerance and autoantibody production. Within each disease, there is considerable heterogeneity in clinical manifestations and disease progression, making diagnosis challenging. Furthermore, treatment options are often limited to general immunosuppressive treatments with significant toxicity and side effects with a limited number of targeted treatments. Due to a lack of predictive biomarkers, treatment decisions are primarily made empirically based on clinical symptoms and limited serological features, such as autoantibodies, resulting in substantial variation in treatment response. Therefore, new clinical strategies, rooted in precision medicine, are needed to accurately predict treatment response, identify novel therapeutic targets, reduce unexplained clinical variation in treatment, and improve clinical outcomes for autoimmune diseases.

Precision is the pursuit of being free from error. Precision medicine is, therefore, the intention to treat each person with as little error as possible using informed and carefully calibrated individually guided therapeutics. Determining the best course of action and moving toward precision medicine in autoimmune diseases entails tailoring targeted therapeutic approaches to an individual based on their underlying disease mechanisms often determined using large-scale molecular profiling and stratification. In some fields, such as oncology, this is already a reality. For example, cancer has a strong genetic component, and next-generation sequencing has led to the extensive use of precision medicine in oncology to aid diagnosis and treatment decisions. Patients with estrogen receptor-positive metastatic breast cancer, for instance, are treated with endocrine therapies (Manohar and Davidson, 2021), whereas patients who express human epidermal growth factor receptor-2 (HER-2) are treated with monoclonal antibodies specifically targeting HER-2 (Goutsouliak et al., 2020). Engineered chimeric antigen receptor (CAR) T cells that recognize specific tumor antigens have also been investigated as targeted individualized therapies for certain blood cancers (Ye et al., 2018). PD-L1 levels are used to determine patients who would benefit from PD-1 antagonists. In addition, precision medicine is used to treat monogenic diseases, such as cystic fibrosis, where affected individuals are treated according to the underlying mutations in the cystic fibrosis transmembrane conductance regulator gene (Lopes-Pacheco, 2020).

Precision medicine in autoimmune diseases has been more challenging due to the complex etiologies of these conditions, involving an interplay between genetic and environmental factors. However, recent technological and bioinformatic advances have helped reveal novel molecular pathways, and characterize disease heterogeneity, leading to the first biopsy-driven clinical trial (Humby et al., 2021), paving the way for precision medicine in autoimmunity. Inspired by the Precision Medicine: Relevance to Autoimmune Disease Colloquium, organized by the Autoimmune Association and Dr. Noel R. Rose in 2020, this review discusses the current understanding of the disease mechanisms, heterogeneity, and pathogenic autoantigens in autoimmune diseases gained from genomic and transcriptomic studies and highlights how these findings can be applied to targeted therapeutic approaches to improve clinical care of patients with autoimmune diseases.

## Resolving patient heterogeneity

Autoimmune diseases are frequently characterized by clinical features or autoantibody prevalence; however, these features are heterogeneous and often overlap between autoimmune diseases, hindering precise diagnosis, and early treatment. Therefore, moving toward molecular diagnostics, which define disease based on changes in biological molecules, may aid diagnosis, and improve clinical outcomes of autoimmune diseases. Recently, exome and genome sequencing have shown promise for identifying pathogenic genetic variants in cases of rare monogenic diseases (Boycott et al., 2019), including patients with autoinflammatory diseases (Kosukcu et al., 2020) such as hereditary fever syndromes. However, these are the rare exceptions. The genetic causes of most autoimmune diseases are complex and genetic risk is determined predominantly by the human leukocyte antigens (HLA) locus, which has the strongest association to rheumatic diseases. Outside of the HLA region, which can account for up to 50% of the genetic risk of a given complex autoimmune trait, hundreds of variants identified through genome-wide association studies (GWASs) each have small additive individual effects, making a diagnosis of autoimmune diseases based solely on genetics currently impossible. In RA, the number of RA-associated risk alleles weighted by the odds ratio correlates with disease risk; however, the predictive power of genetic risk scores is modest and not currently suitable for use in clinical practice (Karlson et al., 2010; Dudbridge, 2013).

As genetic variants identified by GWAS are common variants (generally found in 1% or more of the population – a consequence of study design) and only modestly increase the risk of autoimmune diseases, rare variants with strong effects may contribute to the missing heritability of some patients with autoimmune diseases. For example, following the discovery of mutations in the *TREX1* gene causing the type I interferonopathy Aicardi–Goutières syndrome, *TREX1* variants were identified in up to 0.5–2% of patients with SLE (Lee-Kirsch et al., 2007; Namjou et al., 2011). More recently, exome sequencing identified two rare variants in *BLK* and *BANK1* in a subset of patients with SLE that increased type I interferon (IFN) activity (Jiang et al., 2019). A recently published paper illustrated the role of rare variants in TLR-7 in monogenic SLE demonstrating that with more accessible and available whole exome and genome sequencing, we will learn more about the role of rare variants in autoimmune diseases (Brown et al., 2022). Together, these studies suggest that rare variants may contribute to the genetic risk and clinical heterogeneity of autoimmune diseases. However, the extensive heterogeneity within each autoimmune disease suggests that multiple pathways may contribute to disease; therefore, identifying subgroups of patients with shared molecular signatures is the best avenue to improve the diagnosis and treatment of patients with autoimmune diseases.

As one example, multiple studies have determined subsets of patients with SLE using transcriptomic approaches (Lyons et al., 2012; Banchereau et al., 2016; Toro-Domínguez et al., 2018; Figgett et al., 2019; Panousis et al., 2019; Andreoletti et al., 2021; Sandling et al., 2021). Initial investigations found that approximately half of the patients with SLE exhibit increased peripheral blood expression of type I IFN-regulated genes, termed the "IFN signature," associated with more severe disease (Baechler et al., 2003; Bennett et al., 2003), suggesting that a subset of patients with SLE may benefit from therapies targeting the IFN pathway. Consistent with these findings, the recently approved monoclonal antibody anifrolumab, which targets the type I IFN receptor subunit 1, is effective in about

16% of patients with SLE (Morand et al., 2020). However, there are conflicting reports regarding the effectiveness of stratifying patients based on IFN gene signatures in clinical trials of type I IFN inhibition (Khamashta et al., 2016; Furie et al., 2017; Morand et al., 2020), demonstrating the complexity of the type I IFN response in SLE and identifying the need for additional stratification approaches.

To further refine the IFN signature in SLE, Chiche et al. (2014) found that three distinct transcriptional IFN groups or modules were associated with 87% of patients with SLE and that all types of IFN, not just type I IFN, contributed to the IFN signatures. Importantly, patients with SLE could be further stratified based on the number of active IFN modules (Chiche et al., 2014). In 2016, Banchereau et al. (2016) confirmed and extended these findings in a large cohort of pediatric patients with SLE, identifying overexpression of additional transcriptional modules that correlated with disease activity and clinical parameters of SLE (Banchereau et al., 2016). In addition, patients were stratified into seven clusters based on five immune signatures correlating with disease activity, including type I IFN-, neutrophil-, and plasmablast-associated signatures (Banchereau et al., 2016). Using a similar approach, Toro-Domínguez et al. (2018, 2019) identified three SLE patient clusters characterized by a lymphocyte or neutrophil signature that may respond differently to treatments.

Most transcriptomic studies in SLE use whole blood or bulk cell input making it difficult to discern the affected cell populations. Therefore, single-cell analyses may be necessary to identify and refine molecular clusters in disease-relevant cell state (Perez et al., 2022). Using single-cell RNA sequencing, Nehar-Belaid et al. (2020) defined the cellular subgroups that contributed to the IFN signature in pediatric SLE, including T cells, dendritic cells (DCs), monocytes, and natural killer (NK) cells. Notably, the clustering of these cell types revealed six distinct subgroups of patients associated with disease activity (Nehar-Belaid et al., 2020). In a recent study, Andreoletti et al. (2021) determined unique subgroups of patients based on the transcriptional profiles of sorted monocytes, B cells, CD4+ T cells, and NK cells that correlated with disease activity and ethnicity. In addition, multi-omic approaches may also improve patient stratification, as seen in a study by Guthridge et al. (2020), in which integration of transcriptional modules and autoantibody and soluble mediator profiles identified seven patient clusters with distinct molecular pathways but similar clinical outcomes. In another study, Lanata et al. (2019) used clinical features to define three distinct subgroups of SLE with unsupervised clustering that was supported by differential methylation patterns and ethnicity. Several of these studies explore multi-ethnic cohorts. There are known differences in SLE disease manifestations and severity across different racial and ethnic groups. When exploring biological differences across different patient groups, it's important to note the potential inaccuracy or lack of specification between self-reported and genetic-driven subgroups which may contribute to interpretation problems as ethnicity may be more predictive of differences due to disparities than genetic background (Mersha and Abebe, 2015).

Interestingly, genomic studies have found that autoimmune diseases have shared genetic associations, suggesting that similar pathogenic mechanisms may contribute to different autoimmune diseases (Zhernakova et al., 2009; Richard-Miceli and Criswell, 2012). Indeed, transcriptome and methylome analysis of patients with seven autoimmune diseases demonstrated four patient clusters that differed in the expression of inflammatory, lymphoid, or IFN signature (Barturen et al., 2021). Notably, patients with different autoimmune diseases were found within each cluster (Barturen et al., 2021). Studies using immunophenotyping (Kroef et al., 2020; Martin-Gutierrez et al., 2021) and soluble mediator profiling (Slight-Webb et al., 2021) also found that patients with different autoimmune diseases share similar molecular signatures. Thus, diagnosing patients based on molecular signatures in addition to clinical features may be a key step in moving toward precision medicine and targeted therapeutics.

Taken together, it becomes clear that autoimmune disorders comprise a wide spectrum of clinical manifestations. With the use of genomics, transcriptomics, and other multi-omic approaches, we can begin to examine these complex disorders under a magnifying glass to better define patient heterogeneity and identify targetable genes and pathways.

## Defining the Autoantigenome

As autoimmune disorders are characterized by the body's response to self, defining that exact "self" is critical to both treatment and diagnosis. Autoantibodies, a key component of disease that often directly contribute to outcomes, provide a window into defining these self-antigens and peptides. Of course, autoantibodies do not develop in a vacuum, and certain HLA alleles are strongly associated with autoimmune diseases (Liu et al., 2021), indicating a key role for T cell help and antigen presentation in disease pathogenesis. Understanding and defining the interaction of these three components – autoantibodies, HLA alleles, and T cell repertoire – could identify novel therapeutic targets and molecular diagnostics.

The antibody and T cell repertoires are highly diverse due to recombination of variable, diversity, and joining gene segments, followed by somatic hypermutation in B cell receptors, making the identification of antigen specificity challenging. However, recent advances in Next Generation Sequencing (NGS) and computational approaches have enabled large-scale sequencing of antibody and T Cell Receptor (TCR) repertoires in autoimmune diseases (Zemlin et al., 2002; Schatz and Ji, 2011; Rechavi and Somech, 2017; Nielsen and Boyd, 2019; Nielsen et al., 2019).

Anti-citrullinated protein antibodies (ACPAs) that recognize the posttranslational modification of the amino acid citrulline are a hallmark of RA and contribute to disease pathogenesis (Kurowska et al., 2017). Antibodies consist of two heavy- and light-chain pairs, which both contain antigen-binding domains; therefore, pairing heavy- and light-chains is necessary to determine antigen specificity (Robinson, 2015). To accomplish this, Tan et al. (2014) developed a novel DNA barcoding method to sequence heavy- and light-chain pairs from antibody-producing plasmablasts in ACPA-positive patients with RA and determined affinity-matured clonal families of antibodies. Recombinant expression of 14 antibodies identified four ACPAs with differential targeting of α-enolase, citrullinated fibrinogen, and citrullinated histone H2B (Tan et al., 2014). Additional studies confirmed that ACPAs undergo affinity maturation, resulting in epitope spreading and polyreactivity with other post-translationally modified proteins (Elliott et al., 2018; Titcombe et al., 2018; Kongpachith et al., 2019; Steen et al., 2019).

Repertoire analyses of plasmablasts from healthy individuals with RA-associated autoantibodies demonstrated elevated IgA responses (Kinslow et al., 2016), suggesting that ACPAs may originate from mucosal immune responses. Furthermore, serial analyses of patients with RA found that ACPAs that persisted over time were predominantly IgA (Elliott et al., 2018), consistent with continued mucosal antigen exposure. Therefore, identifying the

specific mucosal antigens targeted by these ACPAs may help identify tolerizing therapies for patients with RA.

Early studies have identified expanded CD4+ T cell clones in the peripheral blood and synovial tissue of patients with RA (Goronzy et al., 1994; Ikeda et al., 1996; Schmidt et al., 1996; VanderBorght et al., 2000; Wagner et al., 2003), including early in the disease course (Klarenbeek et al., 2012). Phenotypic analysis combining TCR sequencing and single-cell transcriptomics revealed expanded memory CD4+ T cell clones with upregulated senescence-related transcripts, chemokine receptors, and CD5 expression, suggestive of antigen stimulation and autoreactivity (Ishigaki et al., 2015). However, the autoantigens targeted by CD4+ T cells in RA remain elusive.

The HLA-DRB1 RA susceptibility alleles contain five shared amino acids of the β1 subunit, referred to as the shared epitope, which is associated with ACPA production (van Gaalen et al., 2004; Huizinga et al., 2005; Busch et al., 2019). There is also significant clinical evidence of differential response based on mechanism in RA patients based on their ACPA/HLA epitope. HLA-DRB1 risk alleles for RA are associated with differential clinical responsiveness to abatacept and adalimumab according to the data from a head-to-head, randomized, single-blind study in autoantibody-positive early RA (Rigby et al., 2021). GWAS analysis demonstrated that an amino acid within the P4 pocket of the peptide-binding groove strongly contributed to the association of HLA-DRB1 and RA (Raychaudhuri et al., 2012), suggesting that the shared epitope may allow binding and presentation of citrullinated autoantigens. Consistent with this hypothesis, antigen discovery analyses using peptide stimulation or peptide–MHC tetramers revealed Th1 and Th17 reactivity to citrullinated antigens, including α-enolase, fibrinogen, vimentin, and aggrecan, in the peripheral blood of patients with RA (Delwig et al., 2010; Law et al., 2012; Scally et al., 2013; James et al., 2014; Gerstner et al., 2020). In addition, T cells specific for citrullinated fibrinogen contribute to the development and progression of RA in mouse models (Hill et al., 2008; Cordova et al., 2013).

Although progress has been made in the identification of autoantigens targeted in autoimmune diseases using microarrays, mass spectrometry, and phage-display assays, these approaches are limited by the need to prespecify the antigens to be studied. Therefore, due to the large number and diversity of antibodies and TCRs, computational methods are needed to predict target antigens from the TCR or antibody sequence alone. Recent progress has been made to predict TCR specificity based on the hypothesis that TCRs that recognize the same antigen share CDR3 sequence motifs. In two separate studies, Dash et al. (2017) and Glanville et al. (2017) developed different algorithms (TCRdist – https://tcrdist3.readthe docs.io/en/latest/ and GLIPH – http://50.255.35.37:8080/, respectively) that clustered TCRs dependent on CDR3 motifs and accurately defined TCR specificity based on these clusters. However, although these approaches are promising, they are limited by the availability of pre-existing knowledge of TCR specificities to make predictions, and large-scale approaches to define these interactions are required. In a recent study, Zhang et al. (2020) clustered tumor TCRs based on antigen-specificity using iSMART and identified novel antigens by integrating TCR clusters, tumor genomics, and HLA genotypes (Zhang et al., 2020). Therefore, multi-omic approaches paired with CDR3 clustering may also help define novel antigens targeted in autoimmune diseases.

In terms of precision medicine, a better understanding of the antigens, TCRs, HLAs, and BCRs driving disease offers a therapeutic window into these diverse disorders. Tolerizing therapies that target specific peptides or regulatory CAR-T cells offer a way to directly suppress autoimmune responses on a patient-by-patient basis.

## The path to targeted therapeutics

Recent genomic and transcriptomic approaches have determined novel pathogenic mechanisms and begun to unravel the heterogeneity of autoimmune diseases, revealing potential therapeutic targets for precision medicine. This section will discuss current work applying knowledge obtained through genomic and transcriptomic studies toward precision medicine approaches.

### Discovering novel therapeutic targets

Genetic analyses of monogenic autoinflammatory diseases have been pivotal in identifying druggable targets that are now used in clinical care (Manthiram et al., 2017). For example, therapies targeting IL-1, such as anakinra, are approved for the inflammasomopathy cryopyrin-associated periodic fever syndrome (Hoffman, 2009). Genetic studies have also revealed efficacious therapeutic targets in polygenic autoimmune disorders. Genetic variation in the Janus kinase family member tyrosine kinase 2 (TYK2), required for type 1 IFN, IL-12 and IL-23 signaling (Sohn et al., 2013; Burke et al., 2019), is associated with autoimmune diseases, including psoriasis (Genetic Analysis of Psoriasis Consortium & the Wellcome Trust Case Control Consortium 2 et al., 2010; Ellinghaus et al., 2012; Tsoi et al., 2012), psoriatic arthritis (Mease et al., 2022), Crohn's disease (Franke et al., 2010), and SLE (Sigurdsson et al., 2005; Graham et al., 2011; Tang et al., 2015; Lee and Bae, 2016). In phase II and III clinical trials, the TYK2 inhibitor deucravacitinib (BMS-986165) was more effective compared to placebo in patients with moderate-to-severe plaque psoriasis (Papp et al., 2018), and is now approved in the US, EU, and other regions. In addition, deucravacitinib is being investigated in early trials of Crohn's disease and SLE demonstrating efficacy in phase II trials in PsA (Mease et al., 2022) and SLE (Morand et al., 2023). However, with polygenic autoimmune diseases, not all identified gene variants may be effective drug targets. Therefore, moving beyond individual genes toward gene networks using *in silico* drug efficacy screening, such as drug–disease network proximity analyses (Kim et al., 2020), to predict potential therapies is needed for drug discovery in autoimmunity. Using this approach, Cordell et al. (2021) recently identified 56 genetic variants associated with primary biliary cholangitis in a genome-wide meta-analysis and predicted several candidate therapies for the disease, including approved treatments of other autoimmune diseases.

Translating genomics to cell function may also identify potentially targetable pathways. Smillie et al. (2019) created a cell atlas of UC using single-cell transcriptomics, highlighting the cells that change in proportions or gene expression compared to healthy tissues. In addition, mapping UC-associated risk alleles onto the cell atlas demonstrated enrichment of risk alleles in individual cell lineages, including M-like cells that exhibited high expression of multiple risk alleles, providing important information about disease etiology and molecular pathways (Smillie et al., 2019). There are several other large-scale efforts to generate single-cell transcriptomic and proteomic datasets in RA and SLE as well as other autoimmune diseases that are able to elucidate cell type specific disease associated genes and pathways (Zhang et al., 2019). However, comprehensive multi-disease cell atlases are needed to provide further insights, which require the integration of multiple large-

scale studies and data sets that may have been collected under different conditions. To avoid confounding variables between studies, multiple computational approaches have been created to remove batch effects (Butler et al., 2018; Haghverdi et al., 2018; Hie et al., 2019; Korsunsky et al., 2019; Polański et al., 2020; Tran et al., 2020). As one example, the algorithm Harmony (Korsunsky et al., 2019) was used to integrate single-cell transcriptomic profiles from multiple disease datasets, revealing a CXCL10+CCL2+ inflammatory macrophage phenotype in the tissues of patients with RA, Crohn's disease, UC, and COVID-19 (Consortium et al., 2021), suggesting that the same pathway may be targeted in distinct diseases.

## Antigen-specific therapies

Identifying antigens targeted by antibodies and T cells in autoimmune diseases will allow for the development of antigen-specific therapies, aiming to restore immune tolerance in autoreactive lymphocytes while maintaining overall immune surveillance to infections and cancer. Tolerogenic DCs have been tested in early phase clinical trials for multiple autoimmune diseases, such as RA, Crohn's disease, and MS (Phillips et al., 2017). In a recent phase 1b trial, autologous tolerogenic DCs loaded with myelin-derived antigens and aquaporin-4 were analyzed for efficacy in MS and neuromyelitis optica spectrum disorders (NMOSDs; Zubizarreta et al., 2019). The tolerogenic DC therapy was well-tolerated and induced IL-10 production by peptide-stimulated cells and a trend toward an increase in regulatory T cells, compatible with tolerance induction (Zubizarreta et al., 2019).

The therapeutic potential of polyclonal regulatory T cells has also been demonstrated in some autoimmune diseases, including MS (Kohm et al., 2002); however, the effects are mostly modest, possibly because of nonspecific regulatory T cells (Raffin et al., 2020). Therefore, based on knowledge acquired from T cell therapies in oncology, another approach is the generation of autologous antigen-specific regulatory T cells by transfecting TCRs or CARs for autoantigens. Kim et al. (2018) transduced human regulatory T cells with a myelin-basic protein-specific TCR isolated from an MS patient and demonstrated that the MBP-specific regulatory T cells suppressed MBP-specific effector cells in vitro and ameliorated disease in a mouse model of MS. Therefore, although still in the preclinical phase, antigen-specific regulatory T cells show promise for the treatment of MS.

## Computational drug repurposing

Identifying new uses for approved drugs, or drug repurposing, will also benefit precision medicine in autoimmune diseases, as conventional drug discovery is often costly and time consuming (DiMasi et al., 2003). However, traditional drug repurposing relies on high-throughput screening technologies that can also be costly; therefore, novel methods are required to expand drug repurposing efforts. Sirota et al. (2011) developed a systematic computational approach to predict disease–drug relationships by comparing gene expression signatures of diseases with those of FDA-approved drugs. This approach identified a novel therapeutic association of an antiepileptic drug, topiramate, with inflammatory bowel disease that was efficacious in a rodent model of colitis (Dudley et al., 2011).

## Caveats and conclusions

Autoimmune disorders are a highly heterogeneous class of conditions. Even within a single clinically diagnosed condition, such as SLE, the underlying causes and manifestations are highly variable. To better treat these disorders and transition toward a precision medicine framework, the last decade of research has used in-depth genetic and genomic studies to better resolve patient heterogeneity and identify the autoantigenome. The progress described in this review represents a substantial leap forward in both our understanding of these complex diseases and their potential treatments.

Although these are not the focus of the current review, there are several additional considerations that are important to note. For complex autoimmune diseases, environment, the interaction of genetics and environment as well as dietary, and lifestyle factors play an important role in affecting disease pathogenesis, progression, and treatment response. For instance, a recent study has shown that oral mucosal breaks trigger anti-citrullinated bacterial and human protein antibody responses in RA demonstrating the role of pathogens and environment in the disease (Brewer et al., 2023). Studies focusing on molecular pathological epidemiology research, which can investigate those factors in relation to molecular pathologies and clinical outcomes have been explored for other conditions such as cancer (Hamada et al., 2017; Hughes et al., 2017; Ogino et al., 2018). It is also important to note that the majority of existing studies in molecular profiling, genomics and genetics of autoimmune diseases have been carried out in patients of European background. If precision medicine is truly the goal, there is a need to explore social determinants of health in the context of disease progression and treatment response in diverse populations. More extensive studies are needed to explore the combination and interaction of the molecular, clinical, social, and environmental factors in diverse patient populations to achieve precision medicine for autoimmunity.

From using genetics to identify new gene targets, to using single-cell genomics to identify cellular and molecular subsets of disease, to computational approaches that aim to merge all this together and repurpose medicine in a targeted fashion, precision medicine in autoimmunity is an endeavor that will continue to yield enormous insights and lead to better – and more importantly – error-free therapeutics.

**Open peer review.** To view the open peer review materials for this article, please visit http://doi.org/10.1017/pcm.2023.14.

**Author contribution.** Y.B. and C.W. wrote the manuscript. M.S. edited the manuscript and oversaw the review. All authors read and approved the final manuscript.

**Financial support.** This work was in part supported by NIH P30 AR070155 (M.S.) and the Autoimmune Association (all authors).

**Competing interest.** M.S. is an advisor to Exagen. M.W. receives research funding from Genentech and Novartis; has received speaking honoraria from Genentech, Novartis, Takeda, and WebMD; and is a co-founder and is on the Board of Directors for Delve Bio. P.I. receives grants support from Intercept and is an advisory board member for Intercept, Advanz, Calliditas, Zydus, Ipsen and CymaBay. F.R.-V. is an employee and stockholder of Bristol Myers Squibb. V.B. is an inventor of patents licensed to Cabaletta Bio related to using engineered T cells for the treatment of autoimmunity.

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
