## [Reviewer Report]

*Comments to Author*: This was a very thorough and encompassing review of the recent literature on the multiple efforts to pave the way for precision medicine in autoimmune diseases.

I have a few minor comments for the author’s considerations.

- It would be relevant to include the recent published paper illustrating the role of rare variants and monogenic SLE in TLR-7 (Vinuesta et al, Nature). Although stated in the paper, with the hopefully more accessible and available WES and WGS, we will learn more of the role of rare variants in autoimmune diseases.

- The AMP has published several papers for precision medicine in RA and SLE that should be included.

- It would be also relevant to include the recent paper of transient bacteremia and ACPAs recently published in Science, highlight the role of pathogens/environment.

- There has also been interesting work in recent scientific meetings of CART cell technology in Antiphospholipid syndrome

Finally, it would be good to see a paragraph acknowledging and highlighting the lack of diversity in molecular profiling, genomics and genetics in autoimmune diseases. If precision medicine is the goal, there is a lot of work to be done in diverse populations.

---

## [Reviewer Report]

*Comments to Author*: The authors wrote a very interesting piece on precision medicine of autoimmune diseases. Autoimmune diseases are very heterogeneous. Especially cautionary aspects of precision medicine are discussed. This is, overall, of very high interest.

The authors discuss big data analyses (such as GWAS) much, but there is relative lack of discussion on environmental influences and gene-by-environment interactions. For complex diseases, actually contributions of environment seem greater than simple mendelian diseases. There are many environmental, dietary, and lifestyle factors that influence diseases, immune system, pathogenic mechanisms.

In line with all of these, for a future direction, it would be worth discussing need of research on dietary / lifestyle / environmental factors, genetics, and personalized molecular biomarkers, all of which are related to precision medicine. The authors should discuss molecular pathological epidemiology research, which can investigate those factors in relation to molecular pathologies and clinical outcomes. Molecular pathological epidemiology has been discussed in Annu Rev Pathol 2019, Curr Colorectal Cancer Rep 2017, J Gastro 2017, etc.